# Bone Sialoprotein Immobilized in Collagen Type I Enhances Angiogenesis In Vitro and In Ovo

**DOI:** 10.3390/polym15041007

**Published:** 2023-02-17

**Authors:** Anja Kriegel, Eva Langendorf, Valentina Kottmann, Peer W. Kämmerer, Franz Paul Armbruster, Nadine Wiesmann-Imilowski, Andreas Baranowski, Erol Gercek, Philipp Drees, Pol Maria Rommens, Ulrike Ritz

**Affiliations:** 1Department of Orthopedics and Traumatology, BiomaTiCS, University Medical Center of the Johannes Gutenberg University, 55131 Mainz, Germany; 2Department of Oral- and Maxillofacial Surgery, BiomaTiCS, University Medical Center Mainz, 55131 Mainz, Germany; 3Immundiagnostik AG, 64673 Bensheim, Germany; 4Department of Otorhinolaryngology, BiomaTiCS, University Medical Center Mainz, 55131 Mainz, Germany

**Keywords:** bone sialoprotein, angiogenesis, chick yolk sac membrane assay, tissue regeneration, osteogenesis

## Abstract

Bone fracture healing is a multistep process, including early immunological reactions, osteogenesis, and as a key factor, angiogenesis. Molecules inducing osteogenesis as well as angiogenesis are rare, but hold promise to be employed in bone tissue engineering. It has been demonstrated that the bone sialoprotein (BSP) can induce bone formation when immobilized in collagen type I, but its effect on angiogenesis still has to be characterized in detail. Therefore, the aim of this study was to analyse the effects of BSP immobilized in a collagen type I gel on angiogenesis. First, in vitro analyses with endothelial cells (HUVECs) were performed detecting enhancing effects of BSP on proliferation and gene expression of endothelial markers. A spheroid model was employed confirming these results. Finally, the inducing impact of BSP-collagen on vascular density was proved in a yolk sac membrane assay. Our results demonstrate that BSP is capable of inducing angiogenesis and confirm that collagen type I is the optimal carrier for this protein. Taking into account former results, and literature showing that BSP also induces osteogenesis, one can hypothesize that BSP couples angiogenesis and osteogenesis, making it a promising molecule to be used in bone tissue regeneration.

## 1. Introduction

In orthopaedics and trauma surgery, various (bio-)materials are frequently applied as osteosyntheses or bone substitutes. Lack of implant osseointegration, implant loosening, and subsequent development of non-unions indicate a need to find new materials that can be applied as implants. The main challenge in this area is the search for a suitable alternative for the gold standard method of autogenous bone grafting in biomaterial research.

Beside developing new materials to improve osseointegration, another approach is to modify (bio-)materials with growth factors or other bioactive molecules. Osteosupportive molecules are, for example: bone morphogenetic proteins (BMPs), collagens or matrix related proteins such as hyaluronic acid [1,2]. In particular, BMPs have been studied intensively, but they cause side effects such as heterotopic ossification or joint stiffness [3,4,5]. It can be concluded that neither the optimal material nor the optimal supportive modification has been found yet. This topic is still a challenge for tissue engineering research.

Bone sialoprotein (BSP), a main component of the non-collagenous part of the extracellular matrix (ECM), plays an interesting role in this context [6]. Like osteopontin, osteonectin, and dentin-sialoprotein, it belongs to the small integrin-binding ligand N-linked glycoprotein (SIBLING) family and is expressed and released by osteoblasts, osteocytes, osteoclasts, odontoblasts, cementoblasts, or hypertropic cartilage cells [7,8]. Structurally, it contains an RGD-binding motif, tyrosine and glutamine acid rich regions, and a collagen binding site [7]. BSP can form a complex with hydroxyapatite (HA), which indicates its role in HA nucleation [9]. Its crucial function in osteogenesis was demonstrated by BSP knock-out mice, which showed impaired bone formation [10,11]. Regarding its diversity, BSP can be considered an ideal candidate for biomaterial functionalisation. Various materials, such as silk, titanium and hydroxyapatite [12,13], have been coated, modified, and treated with BSP to improve their osseointegration; however, the results regarding cell proliferation, differentiation, and gene expression were inconclusive [14,15,16]. Moreover, calcium-phosphate cements coated with BSP demonstrated no enhanced effects regarding bone regeneration in two different rat models [17,18]. These results indicate that the carrier material for BSP might influence its efficiency. One promising candidate to be used as a carrier is collagen as it is the main component of the ECM and has been applied in tissue engineering over recent years [19]. Furthermore, it has been demonstrated that collagen and BSP interact via the collagen binding site [20,21,22]. In a previous study, we were able to show that BSP immobilized in collagen type I induced bone regeneration in vitro and in vivo [23].

Nevertheless, one should bear in mind that bone fracture healing is a multistep process, including early immunological reactions, osteogenesis, and as a key factor, angiogenesis [24]. Without vascularisation no tissue regeneration is possible [25] and many stimulation strategies to enhance vascularization exist [24]. It has been proposed that BSP also plays an important role in angiogenesis. As early as 2000, Bellahcéne showed that BSP mediates endothelial cell adhesion and migration [26] regulated via alpha(V)beta(3) dependent cell adhesion to BSP [27]. Using the BSP-RGD motif in multifunctional protein hydrogels, Mizuguchi et al. showed proangiogenic activity of HUVECs cultured in a three-dimensional cube [28]. In a coculture model of HUVECs and fibroblasts, addition of BSP increased tube formation and vessel formation [29,30]. However, studies regarding the effect of native BSP on angiogenesis in vitro on endothelial cells alone are rare. Therefore, the aim of this study was to analyze the effect of BSP immobilized in a collagen type I gel as carrier for angiogenesis. First, in vitro analyses with endothelial cells (HUVECs) were performed, characterizing proliferation and gene expression of endothelial markers. A spheroid model was employed to detect the effects of sprout formation in a monoculture of HUVECs and a coculture model of HUVECs and human primary osteoblasts. This spheroid model was first described by Korff and Augustin. They developed a three-dimensional spheroid model to prevent apoptosis of endothelial cells as well as a model for endothelial differentiation [31]. During the following years this assay was modified by several groups to investigate angiogenesis in vitro, including cell-cell-interactions, tumour angiogenesis, pathophysiological processes of the capillary system, neo-vessel formation, etc. [32,33]. In the next step, a yolk sac membrane (YSM) assay was performed to analyze the effects of BSP-collagen on vascular formation. This assay was first described by Wang et al. as a novel model to analyse angiogenesis qualitatively as well as quantitatively. They developed this model as an alternative to the well-established chorioallantoic membrane model (CAM) [34]. Compared to the CAM assay, the YSM assay is simpler and more cost effective and is therefore a good tool to test the angiogenic potential of various bioactive molecules or materials [35]. The goal of this study was to detect whether BSP is able to induce angiogenesis in addition to the already detected enhancement of osteogenesis.

## 2. Materials and Methods

### 2.1. Cell Culture

Human umbilical vein endothelial cells (HUVECs) were purchased from Promocell (Heidelberg, Germany) and cultured in complete EBM-2 medium as recommended by the supplier.

Primary human osteoblasts (hOBs) were isolated according to a previously described protocol [36]. Human bone specimens were obtained during hip or knee joint replacement surgeries. The use of residual materials was approved by the ethics committee of the Landesärztekammer Rheinland-Pfalz in agreement with the University Medical Center and in accordance with the principles expressed in the Declaration of Helsinki and the ICH Guidelines for GCP. All patients provided written consent.

### 2.2. Preparation of Collagen Gels (Modified after the Protocol from Wenger [37])

Three-dimensional collagen gels with a concentration of 2.5 mg/mL collagen type I (bovine soluble collagen, Viscofan, Weinheim, Germany) were used and prepared with 50% collagen solution (5 mg/mL), 10% Medium 199 (10×), 6% NaHCO_3_ (7.5%), 2.5% NaOH (1 N), and 31.5% Aqua dest (all from Sigma Aldrich, Steinheim, Germany). BSP (Immundiagnostik, Bensheim, Germany) was added directly into the gels (1 μg/mL and 5 μg/mL BSP) and the amount of Aqua dest. was adjusted accordingly. The cell concentrations used were 5 × 10^5^ HUVEC/mL. One mL of the collagen gel with or without BSP and cells was pipetted into 24 wells for the following experiments. Gelification took place via a temperature change by incubation at 37 °C in an incubator for 20 min.

### 2.3. BSP-Release Assay

All experiments were conducted using human recombinant BSP provided by Immundiagnostik AG (Bensheim, Germany). The recombinant BSP was produced by a stable Chinese hamster ovary (CHO) cell line. As a first step, fluorescein was linked to BSP with the Lightning-Link^®^ conjugation system (Innova Biosciences, Cambridge, UK) according to the manufacturer’s instructions. This fluorescein-linked BSP was incorporated into collagen gels in different concentrations (1 µg/mL, 5 µg/mL, 10 µg/mL, 50 µg/mL and 100 µg/mL). After gelification, the gels were covered with 500 µL/well PBS solution. This solution was exchanged every day, and transferred into a 96-well plate (3 × 100 µL) for direct measuring of the supernatant’s fluorescence intensity (Figure 1A).

### 2.4. Viability Assay

Cell viability of HUVECs in prepared collagen gels (0 μg/mL, 1 μg/mL and 5 μg/mL BSP) was analysed on days 1, 2, 4 and 7 using the alamarBlue^®^ assay (Life Technologies, Karlsruhe, Germany) according to the manufacturer’s instruction. Collagen gels without cells and supplements served as an internal control.

### 2.5. RNA-Isolation/Reverse Transcription/Quantitative Real-Time PCR

According to the viability assay, collagen gels with or without BSP supplementation (1 μg/mL and 5 μg/mL) were prepared. The cell number was adapted to 1 × 10^6^ cells/6 well. After 24 h the gels were digested using a 1 mg/mL collagenase I/dispase solution. The cell suspensions were centrifuged at 1400 rpm for 5 min and the cell pellet was stored at −80 °C until use. Isolation of RNA was conducted with the PeqGold Total RNA Micro Kit (PeqLab) according to manufacturer’s instruction. Total RNA (1 μg) was reverse transcribed into cDNA using dNTPs (4you4 dNTPs Mix (10 mM), BIORON GmbH, Ludwigshafen), Random Primers (Promega, Madison, WI, USA), and MuLV RT (M-MuLV Reverse Transcriptase, M0253S New England Biolabs, Ipswich, MA, USA) according to the manufacturer’s instructions. For gene expression analyses, cDNA template underwent PCR amplification (40 cycles) using the SYBR Green (PowerUp™ SYBR^®^ green master mix, Applied Biosystems, Foster City, CA, USA) and sequence specific primers (Primer sequences listed in Table 1). β2-microglobulin was used for normalization and results were calculated using the well-established 2^−ΔΔCt^ method [38].

### 2.6. Spheroid Model

#### 2.6.1. Spheroid Preparation

Spheroid preparation was performed according to Augustin and Korff 1998 [31]. Six g methocel was dissolved in 500 mL M199 medium (supplemented with 1% L-glut, 1% PS and 10% FCS). The combined solution was stirred overnight at 4 °C and centrifuged at 3500× *g* for 3 h. Meanwhile, the cells were labelled with Cell Tracker™ according to the manufacturer’s instructions (hOBs with Cell Tracker™ Green and HUVECs with Cell Tracker™ Red). One part of the methocel solution (2.4 mL/96-well plate) was diluted with four parts medium (9.6 mL/96-well plate) and the cell suspension (for 1 × 96-well: 6 × 10^4^ cells per mono-culture or 3 × 10^4^ of each cell type per co-culture) was added. After gently mixing the cells, they were seeded in a 96-well plate with U-bottom (100 μL/well corresponding to 500 cells/well) and incubated overnight. On the next day, the spheroids had formed.

#### 2.6.2. Angiogenesis Assay

For angiogenesis assays, the spheroids were embedded in collagen gels composed of 8 parts of collagen, 1 part of M199 10×, and 1 part of NaOH [32].

Spheroids from each 96-well plate were collected with a 1000 μL pipette tip, transferred into a 50 mL tube, and then centrifuged at 300× *g* for 3 min. The supernatant was discarded and the spheroids were gently re-suspended in 1 mL medium M199 + 20% FCS + 0.5% methylcellulose (6 mL methocel solution + 600 μL FCS + 7.8 mL M199 (including 20% FCS)). The same volume of collagen gel (1:1) was added and 1 mL of this mixture was plated in one well of a 24-well plate. Incubation took place at 37 °C. Images were taken via fluorescence microscopy after 1 h and 24 h. Quantification of sprout length was performed with Image J [39] using Fiji distribution and the newest version available. The plugin “angiogenesis” was applied and within this plugin we used the plugin “sprout analysis” according to Eglinger et al. [40].

### 2.7. In Ovo—Yolk Sac Membrane Assay

After cleaning, fertilized Leghorn chicken eggs (LSL Rhein-Main, Dieburg, Germany) were incubated in a special incubator (Janeschitz, Hammelburg, Germany) at a temperature of 37.5 °C and constant humidity. Three days after incubation 5–6 mL of egg clear was collected from the blunt pole, and an oval 3 × 3 cm opening was cut on the upper side of the egg. Subsequently the opening was covered with ParafilmVR (Sigma-Aldrich, St. Louis, MO, USA) to prevent evaporation. The following day, collagen membranes (CM) sized 0.5 × 0.5 cm (Bio-Gide, Geistlich, Baden-Baden, Germany) were inserted under sterile conditions onto the YSM, embryo-distant, near the vessels. The YSM alone was used as negative control group (native). The eggs were further sealed with parafilm and incubated as described above. After 72 h, the vascularization near the CM was photo-documented with a digital microscope at 50- and 100-fold magnification (VHX-1000; Keyence, Neu-Isenburg, Germany). The same region of interest (ROI) of 500 × 500 µm was uniformly applied for every experiment (*n* = 9 per respective CM, in total *n* = 135), the vascular density was measured employing ImageJ.

### 2.8. Statistics

Statistical analyses were performed using the SPSS software (IBM, Version 23) or GraphPad Prism software. The results are presented as medians and quartiles or as means ± standard deviation. Measurements were carried out in triplicates. Cell-based experiments were independently repeated three times. Normally distributed data were analysed by one-way ANOVA. Depending on Levene’s test for equality of variances, pairwise comparisons were conducted either by a Tukey-HSD or Games-Howell post hoc test. In contrast, non-normally distributed data were evaluated with the Kruskal–Wallis test. For pairwise comparisons, the Mann–Whitney U-test was used. *p* < 0.05 was considered statistically significant (* *p* < 0.05, ** *p* < 0.01, *** *p* < 0.005, and **** *p* < 0.001). Due to multiple testing, the *p*-values were adjusted through the Bonferroni–Holm method.

## 3. Results and Discussion

### 3.1. BSP Immobilized in Collagen Type I Enhances Proliferation of HUVECs

As a first step, the release of BSP had to be determined. We were able to demonstrate that BSP is released from the collagen gels in a constant manner. Within 7 days, approximately 80% was liberated (Figure 1).

To our knowledge, this is the first study analysing the release of BSP from collagen gels. Collagen is a commonly used carrier for bioactive molecules and the shown release kinetics are comparable with kinetics from other bioactive molecules [41]. BSP is released to the extracellular matrix (ECM) by osteoblasts and as collagen is the main component of the ECM it offers itself as a carrier material [42]. The hypothesis that collagen might be the optimal carrier for BSP was first proposed by Kruger et al. who demonstrated that collagen type I and BSP interact with each other via a collagen binding site of BSP [21]. Regarding bone regeneration it has been demonstrated that BSP immobilized in and released from collagen type I shows a positive effect on osteoblast differentiation and bone repair [23,43]. An inducing effect on mineralization was observed for a complex built from collagen and BSP [22].

HUVECs were seeded in collagen type I gels with and without BSP. Figure 2 demonstrates that the cells are well incorporated into the gel and demonstrate a typical morphological structure. At first view, no differences can be observed whether BSP is added or not. This is in accordance to our former results, when HUVECs were incorporated into the gel in a coculture model with osteoblasts [23].

A few studies exist which analyze the effect of BSP on endothelial cells, but it is known that BSP binds to cells via integrin alpha_5_beta_3_ and induces migration of endothelial cells [27]. Most likely the RGD motif is involved as this motif promotes the proangiogenic activity of HUVECs [28].

Figure 3A shows the effect of immobilized BSP, in a three-dimensional collagen gel, on the viability and proliferation of endothelial cells. The most significant effect was observed after four days of culture, when in particular, the group with the low BSP concentration showed an enhanced expression of endothelial markers compared to the control without BSP. After seven days the effects were not as pronounced; however, proliferation of HUVECs in both BSP groups was significantly enhanced compared to the control group.

There are few studies which analyze the effect of BSP on endothelial cells. Interestingly, most already observed significant effects with BSP concentrations between 0.3 and 3 µg/mL. Jain et al. observed that BSP binds MMP-2 and induces vessel formation [29]. Byzova et al. and Bellecene et al. demonstrated a positive effect of these low concentrations on HUVEC adhesion [26,27]. This is in accordance with our results and speaks for a good BSP effect even with low concentrations. Moreover, this is confirmed by BSP studies with osteogenic cells: proliferation of human primary osteoblasts on calcium phosphate scaffolds was enhanced after BSP-coating with a low concentration [15]. Baht et al. demonstrated a concentration dependent binding curve of osteoblasts with a saturation at a concentration of 200 nM approximately corresponding to the used concentration of 5 µg in this study [44].

Regarding economic aspects, this fact makes BSP an even more interesting molecule for medical applications.

### 3.2. Effect of BSP Encapsulation in Collagen Gels on Gene Expression in Endothelial Cells

BSP addition enhanced gene expression of the endothelial and other pro-angiogenic markers, namely KDR (VEGF receptor 2), PECAM (platelet/endothelial cell adhesion molecule-1), IGF-1 (insulin-like growth factor 1), VEGF (vascular endothelial growth factor), and to a smaller extent MCAM (melanoma cell adhesion molecule) compared to untreated control (Figure 2B). The highest effect was seen in the higher BSP concentration (5µg/mL). No effects were observed regarding the gene expression of vWF.

To our knowledge, the effect of bone sialoprotein on gene expression of endothelial markers has not been analysed before. Our results show that the typical endothelial markers are upregulated in collagen gels with immobilized BSP. The most pronounced effect was observed in the gene expression of IGF-1 with a more than 5-fold enhancement. IGF-1 plays a role in angiogenesis [45] as well as in osteogenesis [46]. It has been demonstrated that BSP and IGF-1 expression are coupled. BSP −/− mice demonstrate a decreased gene expression level of IGF-1 [47]. Upon administration of IGF-1 in dental pulp stem cells BSP gene expression is upregulated [48]. In addition, it has been shown that BSP expression is downstream of IGF-1 [49,50]. Nevertheless, the exact mechanisms still have to be elucidated. In summary, the results of gene expression support the hypothesis that BSP plays a role in angiogenesis.

### 3.3. BSP Induces Sprout Number and Length in a Spheroid Model

In order to characterize the effect of BSP on vascular formation, a spheroid model was employed and the sprout formation was analysed. In this three-dimensional model, no sprout formation could be observed for HUVECs, when cultured as a monoculture (data not shown). This is in accordance with former studies, where a monoculture of endothelial cells was not able to form sprouts or vessels and additional factors were needed to induce these effects [36,51]. One approach to induce vessel formation is a coculture with other cells, for example, osteoblasts. It has been well described that these cells influence each other and enhance osteogenesis as well as angiogenesis [23,52,53]. Therefore, the spheroid model experiment was performed with a coculture of osteoblasts and endothelial cells.

Figure 4A shows the experimental design of the spheroid model. The number of sprouts/spheroids (Figure 4B) as well as cumulative sprout lengths (Figure 4D) are significantly enhanced in both BSP groups compared to the coculture without BSP. No differences were observed in individual sprout length (Figure 4C) and sprout diameter (Figure 4E) when compared to control. Figure 4F,G displays the embedded spheroids after gelification (1 h) and an exemplary picture of each group (green—osteoblasts, red—endothelial cells) after 24 h, respectively.

BSP, particularly the concentration of 5 µg/mL, significantly enhanced sprout number and length. The most prominent effects were observed regarding the gene expression for this same concentration. The positive effect on neo-vascularization is in accordance with Bellahcéne et al. who were the first to describe pro-angiogenic effects of BSP by demonstrating that BSP enhances the proliferation and migration of HUVECs [26]. Only two studies exist—one for a breast cancer cell line, another with osteosarcoma cells in coculture with adipocyte stem cells—showing that in a 3D-hydrogel spheroid model, the gene expression of BSP was enhanced [50,54]. The effect of BSP on angiogenesis in spheroid models has not been analysed so far. Some studies exist which analyze the effect of different BSP concentrations on osteogenic markers. In these studies, concentrations between 1 and 10 µg/mL demonstrated the highest impact [23,55,56].

### 3.4. BSP Enhances Vascular Density in the YSM-Assay

In order to assess the angiogenic potential of BSP in combination with collagen type I, a yolk sac membrane assay was performed. BSP was adsorbed to the collagen membrane in a defined concentration, laid on the yolk sac membrane of fertilized chicken eggs and the vascular density was measured after 72 h.

Figure 5A–C shows exemplary pictures of the collagen membrane without (5A), with 0.5 µg (5B) or 5 µg BSP (5C). The arrow indicates a vascular connection to the membrane in the group with the highest BSP concentration. Quantification of vascular density shows that both BSP groups enhance the density significantly, especially the group loaded with 5 µg BSP.

Bellahcéne et al. demonstrated that BSP alone was able to promote angiogenesis in the CAM-assay [26,47]. Our results show that BSP immobilized in collagen type I is released as a still bioactive molecule and induces angiogenesis.

The crosstalk between angiogenesis and osteogenesis is essential for bone tissue regeneration [57] and BSP might be one component of this crosstalk.

## 4. Conclusions

Our results demonstrate that BSP is capable of inducing angiogenesis, interestingly, at lower rather than higher concentrations. This should be further analysed in follow-up studies, especially the mechanisms concerning why higher concentrations do not result in better angiogenic effects as this represents an important issue. Moreover, our results confirm that collagen type I is the optimal carrier for BSP. Nevertheless, it might be interesting to test other collagen types as alternatives.

Taking into account former results, and literature showing that BSP also induces osteogenesis, one can hypothesize that BSP couples angiogenesis and osteogenesis, making it a promising molecule to be used in bone tissue regeneration.

## Figures and Tables

**Figure 1 polymers-15-01007-f001:**
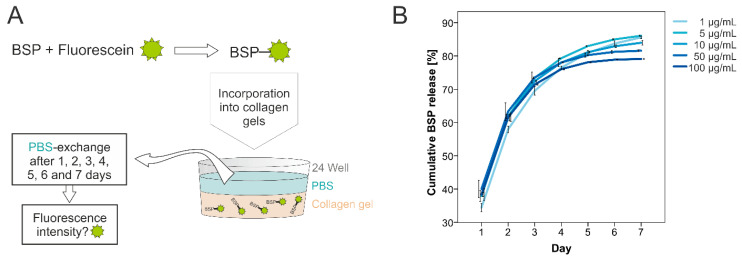
BSP release. (**A**)—Fluorescein-coupled BSP was incorporated into collagen gels. The gels were covered with PBS solution that was exchanged every day, followed by measuring the solutions’ fluorescence intensity. (**B**)—Cumulative BSP release from collagen type I gels. Different concentrations were used for the release assays (1–100 µg/mL). The diagram shows the BSP release presented as cumulative percentage (*n* = 3 per concentration).

**Figure 2 polymers-15-01007-f002:**
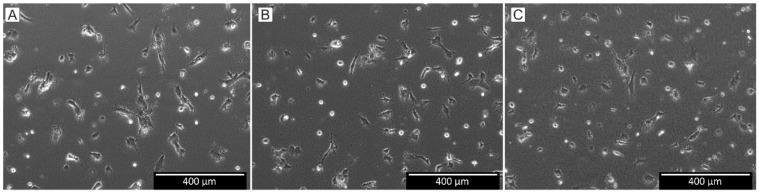
HUVECs seeded in three-dimensional collagen type I gels with different BSP concentrations. (**A**)—0 BSP; (**B**)—1 µg/mL BSP; (**C**)—5 µg/mL BSP.

**Figure 3 polymers-15-01007-f003:**
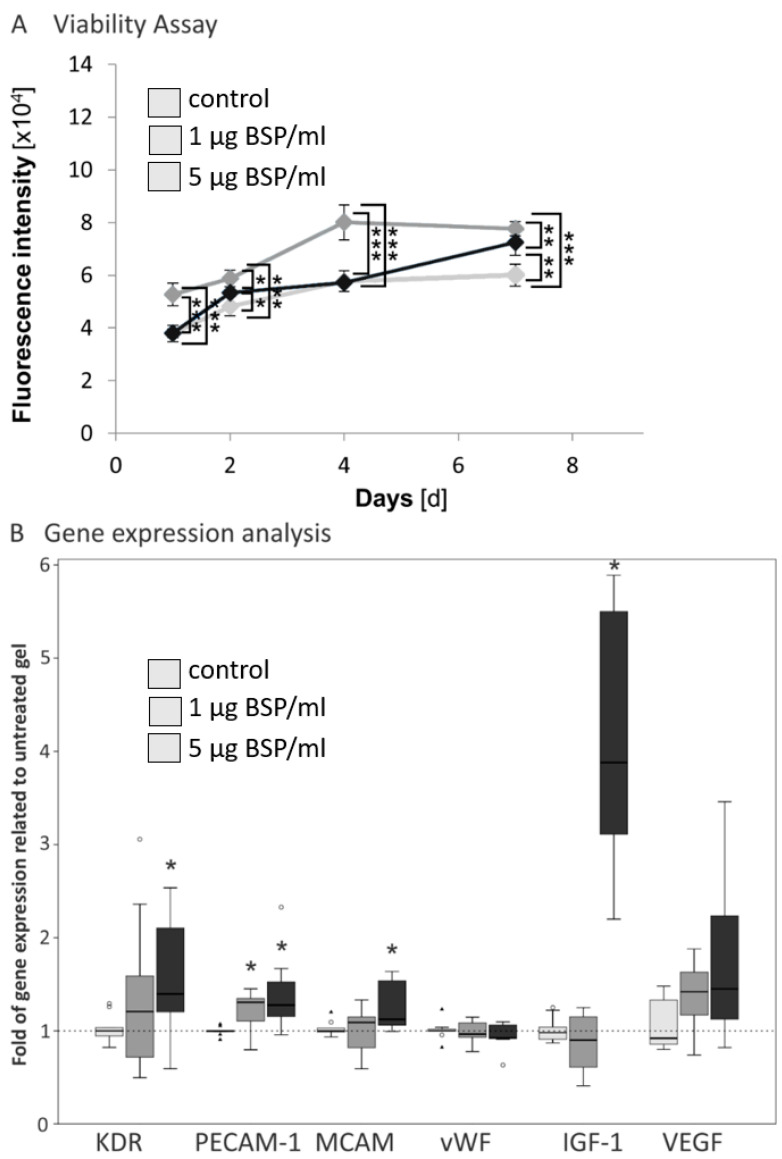
Cell viability and proliferation assay and gene expression analysis. (**A**)—Cell viability of HUVECs in collagen gels with immobilized BSP. Results are expressed as mean ± SD (*n* = 9). Games-Howell or Tukey-HSD post hoc tests (dependent on Levene’s test) revealed significant differences (** *p* < 0.01, *** *p* < 0.005). (**B**)—Relative gene expression analyses in endothelial cells seeded in different modified collagen gels compared with untreated collagen gels (gene expression = 1). Results are expressed as median and quartiles (*n* = 9). Mann–Whitney-U tests revealed significant differences (* *p* < 0.05, circles and triangles present outliners).

**Figure 4 polymers-15-01007-f004:**
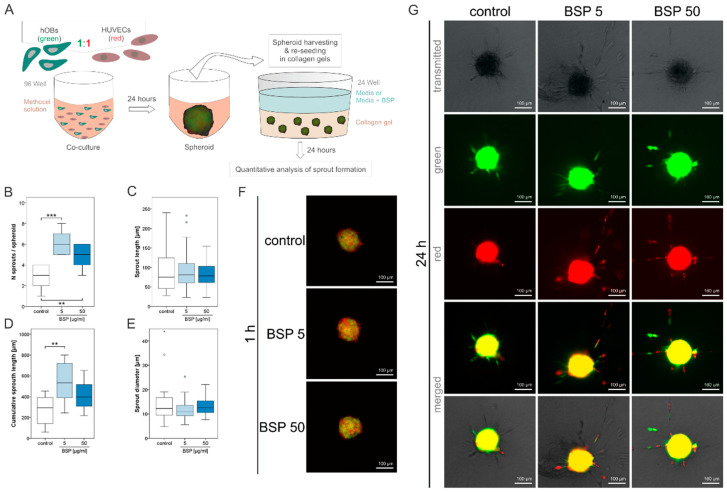
Spheroid model. (**A**)—Methodological overview of the spheroid model. (**B**)—Number of sprouts counted per spheroid. (**C**)—Sprout length. (**D**)—Cumulative sprout length. (**E**)—Sprout diameter, significant differences are indicated (* *p* < 0.05, ** *p* < 0.01, *** *p* < 0.005). (**F**)—Spheroids embedded in collagen gels after 1 h. (**G**)—Spheroids embedded in collagen gels after 24 h.

**Figure 5 polymers-15-01007-f005:**
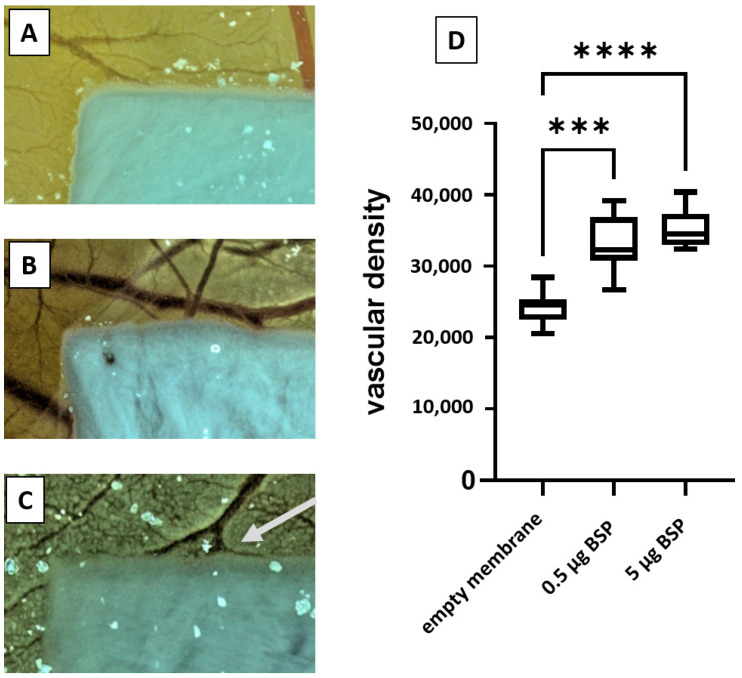
Yolk sac membrane assay. Collagen membrane without BSP (**A**), with 0.5 µg BSP (**B**) or 5 µg BSP (**C**) 72 h after putting the membrane onto the yolk sac membrane. (**D**)—Quantitative analyses of vascular density. (*** *p* < 0.005; **** *p* < 0.001).

**Table 1 polymers-15-01007-t001:** Primer sequences.

Gen	Forward Primer	Reverse Primer
b_2_-microglobulin	CTC ACG TCA TCC AGC AGA GA	ACG GCA GGC ATA CTC ATC TT
IGF-1	CCT GAC CTT GTG ATT TGC CC	TCC CCT TGA AAG ACC CCA TC
KDR	TTA CTT GCA GGG GAC AGA GG	TTC CCG GTA GAA GCA CTT GT
MCAM	CGG CAA GTG AAC AAG ACC AA	GTC TGG TGT GAG GGT GGT TA
PECAM	CAT TGG CGT GTT GGG AAG AA	GCT CAT GTT TGC CTA GCT CC
VEGF	AGA TGA GCT TCC TAC AGC ACA AC	AGG ACT TAT ACC GGG ATT TCT TG
vWF	GGA TTC AGT GGA TGC AGC AG	TAG GGA GGT CTT CGA TTC GC

## Data Availability

Data available on request due to restrictions. The data presented in this study are available on request from the corresponding author.

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
