# Peer review of "Bone Sialoprotein Immobilized in Collagen Type I Enhances Angiogenesis In Vitro and In Ovo"

_polymers, 2023, doi:10.3390/polym15041007_

Round 1

Reviewer 1 Report

This study aimed to use BSP stable in Col-1 to observe the angiogenesis from in vitro (i.e, HUVECs proliferation, gene expression, and spheroid model) and in ova (yolk sac membrane assay). Authors found out that Col-1  can be a good carrier for BSP to induce both osteogenesis and angiogenesis.  Overall, I knew this is continued work after the authors published the previous work Kriegel, Anja, et al. International Journal of Bioprinting 8.3 (2022). Please check the following points before publication.

- What’s your finding about the BSP concentration difference in Fig. 2?  Can you add an additional picture for BSP 50 µg/ml ? 

- The author should describe and discuss the results of Fig. 3 A, not just compare BSP with control group.  Why 1 µg/ml is better than 5 µg/ml (with a significant difference compared each other on day 2, 4, 7)? How about comparing BSP 50 µg/ml ? 

- Did author gene expression of  Fibroblast growth factors (FGF) in Fig. 3 B  ? And why did the author measure vWF for this experiment? 

- Lack of BSP 1 µg/ml compared to other BSP 5 µg/ml and BSP 50 µg/ml groups. Why BSP 50 µg/m didn’t have the highest impact ?

- How author measure the sprout length  and sprout diameter from spheroid model ? It should be method in 2. Materials and Methods

- Why author design BSP 0.5 µg/ml instead of 1 µg/ml in FIg. 5 from Yolk sac membrane assay? 

- I can’t see your supplementary materials, including Figure S1: title; Table S1: title; Video S1: title.  ????

Author Response

Reviewer 1:

This study aimed to use BSP stable in Col-1 to observe the angiogenesis from in vitro (i.e, HUVECs proliferation, gene expression, and spheroid model) and in ova (yolk sac membrane assay). Authors found out that Col-1  can be a good carrier for BSP to induce both osteogenesis and angiogenesis.  Overall, I knew this is continued work after the authors published the previous work Kriegel, Anja, et al. International Journal of Bioprinting 8.3 (2022). Please check the following points before publication.

  1. What’s your finding about the BSP concentration difference in Fig. 2?  Can you add an additional picture for BSP 50 µg/ml ? 

Answer: We thank the reviewer for this comment as he/she is right and the paragraph is a bit misunderstanding. We rewrote the paragraph as follows:

HUVECs were seeded in collagen type I gels with and without BSP. Figure 2 demonstrates that the cells are well incorporated into the gel and demonstrate a typical morphological structure. On a first view, no differences can be observed whether BSP is added or not. This is in accordance to our former results, when HUVECs were incorporated into the gel in a coculture model with osteoblasts (Kriegel et al, 2022).

Analogue to our former experiments we only tested the BSP concentrations of 1 and 5 µg. Only the release assay was performed with higher concentrations up to 100 µg/ml to show that the release is constant and independent from the BSP concentrations. We and others could demonstrate that rather low concentrations demonstrate better effects (see next answer to reviewer comment)

  1. The author should describe and discuss the results of Fig. 3 A, not just compare BSP with control group.  Why 1 µg/ml is better than 5 µg/ml (with a significant difference compared each other on day 2, 4, 7)? How about comparing BSP 50 µg/ml ? 

Answer: We agree with the reviewer and added a few points to the discussion:

Only few studies exist analyzing the effect of BSP on endothelial cells. Interestingly most observed already significant effects with BSP-concentrations between 0,3 and 3 µg/ml. Jain et al. 2008 observed that BSP binds MMP-2 and induced vessel formation.  Byzova 2000 and Bellecene et al. 2000 demonstrated a positive effect of these low concentrations on HUVEC adhesion. This is in accordance to our results and speaks for a good BSP effect even with low concentrations. Moreover this is confirmed by BSP studies with osteogenic cells: proliferation of human primary osteoblasts on calcium phosphate scaffolds was enhanced after BSP-coating with a low concentration (Klein/Baranowski 2018). Regarding economic aspects, this fact makes BSP an even more interesting molecule for medical applications.

  1. Did author gene expression of  Fibroblast growth factors (FGF) in Fig. 3 B  ? And why did the author measure vWF for this experiment? 

Answer: We thank the reviewer for this hint. We have not analyzed FGF gene expression, but will consider it for our future experiments.

We analyzed vWF as this factor has often been analyzed as angiogenic marker in HUVECs in combination with PECAM, VEGF and KDR (for example Inglis et al., 2016, DOI: 10.1186/s13287-015-0270-3 and Shimazaki et al., 2021, doi: 10.3390/biology10111212)

  1. Lack of BSP 1 µg/ml compared to other BSP 5 µg/ml and BSP 50 µg/ml groups. Why BSP 50 µg/m didn’t have the highest impact ?

Answer: We thank the reviewer for pointing out this fact. Actually we cannot explain, why BSP 50 µg/ml does not have the highest impact. We can only refer to our former experiments as well to studies performed by other groups (see answer before).

Regarding osteoblasts Baht et al. demonstrated a concentration dependent binding curve, with a saturation at a concentration of 200 nM approximately corresponding to the used concentration of 5 µg in this study.

We added a respective sentence to the manuscript.

  1. How author measure the sprout length  and sprout diameter from spheroid model ? It should be method in 2. Materials and Methods

Answer: We thank the reviewer for this important comment and added a more detailed description to the Material and Method section as well as one further literature, where the method is described in detail (DOI: 10.1186/s41232-016-0033-2).

  1. Why author design BSP 0.5 µg/ml instead of 1 µg/ml in FIg. 5 from Yolk sac membrane assay? 

Answer: We agree with the reviewer that on a first view this seems to be not consequent. However, the Yolk sac membrane assay was performed after the in vivo experiments described in our former publication (Kriegel et al., 2022) and the concentrations were chosen as absolute amount based on this experimental set up.

  1. I can’t see your supplementary materials, including Figure S1: title; Table S1: title; Video S1: title.  ????

Answer: Thanks for your careful reading. We only submitted a table with the primer sequences, neither a figure nor a video. The respective words are now deleted in the text. 

Reviewer 2 Report

An interesting topic was studied by authors, this paper was well-organized. I have some suggestions that might be helpful to improve the paper, after then, I suggest the acceptance of this paper. My comments as follows: 

1) The spheroid model and yolk sac membrane assay used in this paper should be emphasized in this introduction section as both methods are very important for the whole paper. 

2) The information of BSP should be given in 2.3 subsection. 

3) Line 146, it should be Figure 1A instead of Figure 3A, please confirm. 

4) There was no Figure 4 in the paper. 

5) In figure 2, I suggest the authors could provide pictures with different magnifications on different days. 

6) I suggest the authors should add more discussion about why collagen type I is optimal for carrying the BSP, how about other polymers gels? 

7) The conclusion section could be further improved. 

Author Response

Reviewer 2:

An interesting topic was studied by authors, this paper was well-organized. I have some suggestions that might be helpful to improve the paper, after then, I suggest the acceptance of this paper. My comments as follows: 

  • The spheroid model and yolk sac membrane assay used in this paper should be emphasized in this introduction section as both methods are very important for the whole paper. 

Answer: We agree with the reviewer and added the following sentences to the introduction. Thanks for this advice:

This spheroid model has first been described by Korff and Augustin. They developed a three-dimensional spheroid model to prevent apoptosis of endothelial cells as well as a model for endothelial differentiation[31]. During the next years this assay has been modified by several groups to investigate angiogenesis in vitro including cell-cell-interactions, tumor angiogenesis, pathophysiological processes of the capillary system, neo-vessel formation, etc. [32,33]. In the next step, a yolk sac membrane (YSM) assay was performed to analyze the effects of BSP-collagen on vascular formation. This assay has first been described by Wang et al. as a novel model to analyse angiogenesis qualitatively as well as quantitavely. They developed this model as an alternative to the well established chorioallantoic membrane model (CAM)[34]. Compared to the CAM assay the YSM assay is more simple and cost effective and is therefore a good tool to test the angiogenic potential of various bioactive molecules or materials[35].

  • The information of BSP should be given in 2.3 subsection.

Answer: We added the information to subsection 2.3 – thanks for your careful reading.

  • Line 146, it should be Figure 1A instead of Figure 3A, please confirm. 
  • There was no Figure 4 in the paper. 

Answer: We apologize; somehow the figures got mixed up – we corrected the numbering of the figures.

  • In figure 2, I suggest the authors could provide pictures with different magnifications on different days. 

Answer: Although we agree with the reviewer we have to apologize that we do not have other magnifications or pictures taken on other days. We just wanted to have an overview on the first day after seeding that the cells are incorporated in the gel and show a typical phenotype. In further studies we will take pictures on the following days with different magnifications – thanks for the hint!

  • I suggest the authors should add more discussion about why collagen type I is optimal for carrying the BSP, how about other polymers gels? 

Answer: As suggested by the reviewer, we formulated in more detail, why we believe that collagen type I is a good carrier for BSP:

To our knowledge, this is the first study analyzing the release of BSP from collagen gels. Collagen is a commonly used carrier for bioactive molecules and the shown release kinetics are comparable with kinetics from other bioactive molecules [41]. BSP is a released to the extracellular matrix (ECM) by osteoblasts and as collagen is the main component of the ECM it offers itself as carrier material[42]. The hypothesis that collagen might be the optimal carrier for BSP has been proposed first by Kruger et al. who demonstrated that collagen type I and BSP interact with each other via a collagen binding site of BSP [21]. Regarding bone regeneration it has been demonstrated that BSP immobilized in and released from collagen type I shows positive effects on osteoblast differentiation and bone repair [23,43]. An inducing effect on mineralization was observed for a complex built from collagen and BSP[22].

Regarding the question, whether other polymers might be good carriers we added a respective sentence into the conclusion and outlook section. Especially other collagen types might be good alternatives and should be analyzed.

  • The conclusion section could be further improved. 

Answer: As suggested by the reviewer we improved the conclusion section:

Our results demonstrate that BSP is capable to induce angiogenesis, interestingly rather at lower than higher concentrations. This should be further analysed in follow-up studies; especially the mechanisms why higher concentrations do not result in better angiogenic effects represents an important issue. Moreover, our results confirm that collagen type I is the optimal carrier for BSP. Nevertheless, it might be interesting to test other collagen types as alternatives.

Taking into account former results and literature showing that BSP also induces osteogenesis one can hypothesize that BSP couples angiogenesis and osteogenesis making it a promising molecule to be used in bone tissue regeneration.

Round 2

Reviewer 1 Report

Authors already addressed my questions and make a new revision version. I suggest to accept for publication. 

Reviewer 2 Report

I suggest the paper can be accepted in the current version. Please revise "Conclusion and Outlook" to "Conclusion".